# Early Detection of Macular Atrophy Automated Through 2D and 3D Unet Deep Learning

**DOI:** 10.3390/bioengineering11121191

**Published:** 2024-11-25

**Authors:** Wei Wei, Radhika Pooja Patel, Ivan Laponogov, Maria Francesca Cordeiro, Kirill Veselkov

**Affiliations:** 1Department of Surgery and Cancer, Imperial College London, London SW7 2AZ, UK; w.wei20@imperial.ac.uk (W.W.); radhika.patel6@nhs.net (R.P.P.); i.laponogov@imperial.ac.uk (I.L.); 2Imperial College Ophthalmology Research Group, London NW1 5QH, UK; 3Ningbo Medical Center Lihuili Hospital, Ningbo 315040, China

**Keywords:** macular atrophy, deep learning, optical coherence tomography

## Abstract

Macular atrophy (MA) is an irreversible endpoint of age-related macular degeneration (AMD), which is the leading cause of blindness in the world. Early detection is therefore an unmet need. We have developed a novel automated method to identify MA in patients undergoing follow-up with optical coherence tomography (OCT) for AMD based on the combination of 2D and 3D Unet architecture. Our automated detection of MA relies on specific structural changes in OCT, including six established atrophy-associated lesions. Using 1241 volumetric OCTs from 125 eyes (89 patients), the performance of this combination Unet architecture is extremely encouraging, with a mean dice similarity coefficient score of 0.90 ± 0.14 and a mean F1 score of 0.89 ± 0.14. These promising results have indicated superiority when compared to human graders, with a mean similarity of 0.71 ± 0.27. We believe this deep learning-aided tool would be useful to monitor patients with AMD, enabling the early detection of MA and supporting clinical decisions.

## 1. Introduction 

Age-related macular degeneration (AMD), a leading cause of visual impairment and irreversible blindness among the elderly population in developed countries, poses a significant public health challenge. Projections estimate the number of affected individuals will rise to 288 million by 2040 [1], accounting for approximately 8.7% of global blindness cases [2]. AMD progresses through stages, initially characterized by drusen deposits and retinal pigment epithelium (RPE) abnormalities and eventually leading to choroidal neovascularization (CNV) and macular atrophy (MA). It can be classified into early, intermediate, and late stages, based on a clinical classification system [3].

Although AMD may present at different stages, the endpoint of AMD is MA, which is irreversible and characterized by the permanent loss of photoreceptors, RPE, and the underlying choriocapillaris [4]. MA can be seen in both advanced dry and wet AMD (representing as neovascularization) [5,6]. In addition, the conversion of MA from advanced wet AMD is a common phenomenon, with a high incidence seen in prominent clinical trials of wet AMD, varying from 30–98% depending on the length of the follow-up [5,7,8,9,10,11]. New treatments are emerging, with two recently having been approved by the FDA in 2023 [12,13,14]; however, they are only available in the USA and are not widely used so far because of the high prices. Therefore, detecting MA in both dry and wet AMD is still essential and urgent for monitoring the progression of AMD globally.

In 2018, a new consensus to define MA using optical coherence tomography (OCT) was published [15]. The classification has shown that atrophy undergoes different stages of evolution, with four histological OCT features proposed: complete RPE and outer retinal atrophy (cRORA), incomplete RPE and outer retinal atrophy (iRORA), complete outer retinal atrophy (cORA), and incomplete outer retinal atrophy (iORA) [15]. The criteria are based on the absence or interruption of RPE and photoreceptors and whether the region of hypertransmission is less or more than 250 μm in diameter [15,16,17]. This consensus provides a standard to assess MA at an early stage based on OCT rather than other conventional imaging modalities, including color fundus photography (CFP) and fundus autofluorescence (FAF). 

Recently, the automated segmentation of medical images has become a highly and intensively focused research topic, as manual annotation is time-consuming when faced with large volumes of medical data [18]. Therefore, a fully automated segmentation process is highly desirable [19]. However, the biggest challenge is still the high variability in the lesion location, shape, margin, and size, together with the image quality in the real world, which often compounds the disagreement between human graders. The Unet algorithm, based on a state-of-the-art fully convolutional algorithm, is one of the most popular deep learning networks for medical semantic segmentation and has achieved encouraging performances because of its strong ability to detect lesion boundaries. Its 3D variant, 3D Unet, inspired by the famous 2D Unet [20], extends these capabilities to volumetric data. It has shown superior performance compared to 2D Unet in some areas [21].

Despite these advances in OCT segmentation, there is still an urgent need for detecting MA at an early stage in patients with AMD, before irreversible visual loss occurs. However, there remains a lack of tools to detect MA at early stages, as we have previously described [22], with most being focused on the late stages of MA using 2D Unet [23,24,25]. Nevertheless, OCT provides a three-dimensional retinal structure through volumetric scans, opening the door to more detailed assessments. Moreover, the recent 3D Unet can more accurately detect lesions in three-dimensional space compared to 2D Unet, which could further improve the accuracy of the model’s performance. Previously, we have developed a 2D CNN model based on Unet architecture which successfully detected atrophy quantitatively using OCT in patients with AMD [22]. We hypothesized that 3D analysis would provide greater accuracy in detecting MA. In this study, we aimed to develop a 3D Unet-based architecture to detect atrophic-associated features based on OCT volumetric images and further compare the performance of 2D and 3D Unet-based segmentation.

## 2. Materials and Methods

### 2.1. Ethics and Clinical Research

This retrospective and observational study was conducted on a database based at Western Eye Hospital, London, and approved by the HRA ethics committee (IRAS project ID: 291183; REC reference: 21/HRA/0033). The database was called OCTANE (“Pilot OCT/A National Enterprise Study”). All patients in OCTANE had been diagnosed with AMD, either at an early, intermediate, or late stage, during the period from 2000 to 2021, with a maximum follow-up period of 5 years. These patients regularly attended the Macula Clinic and were being followed up for AMD. All patients’ data were fully anonymized by stripping all identifiable medical information, e.g., patient’s name, DOB, address, etc. The clinical study was conducted in accordance with the World Medical Association’s Declaration of Helsinki and other relevant regulations. 

### 2.2. Data Collection and Image Processing

For SD-OCT scans, an OCT volume of the central 20° × 20° centered on the fovea was acquired with 49 B scans in each volume using a Spectralis HRA + OCT device (Heidelberg Engineering GmbH, Heidelberg, Germany) in regular real-world clinics, with a resolution of 512 × 498 pixels in each B scan. The built-in follow-up mode was used to reacquire follow-up volumetric scans after the baseline. 

Exclusion criteria consisted of (1) no MA during the whole follow up, (2) images with low resolution (quality), high noise, and average quality < 18 in Spectralis, (3) incorrect imaging modality, 25 or 61 slices at a time, and (4) other eye conditions that affect image capturing, like corneal mild haze, mild subcapsular cataract, and vitreous degenerative conditions.

Follow-up volumetric OCTs from baseline were acquired from 125 eyes (89 patients), which were totally from the real world. Amongst these, 77 eyes had a diagnosis of dry AMD and 48 eyes had wet AMD. In total, 1241 volumetric OCTs were collected, including 319 volumes from dry AMD and 922 volumes from wet AMD. Each volume included 49 B-scans.

### 2.3. Manual Annotation

From the accumulated volumetric OCTs, the last follow-up images from all 125 eyes (89 patients) with MA were collected for manual annotation, making a total of 4466 raw B scan images.

All 4466 anonymized images were manually annotated by two experienced graders (WW and RPP) who had received systematic annotation training with standardized labeling criteria based on atrophy criteria [15] in Labelme (an open-source annotation software). For training and validating datasets, two graders annotated separately, and when a disagreement of annotation occurred between the two graders they had a discussion and re-annotated together. After that, these annotations made by the two graders in training and validating datasets were used as the ground truth. For the testing dataset, the two graders annotated independently, and the results were compared in this study.

Detailed annotations with structural and morphological changes in OCT included six atrophic-associated labels: interruption of the outer retina, interruption of RPE, absence of the outer retina, absence of RPE, hypertransmission < 250 μm, and hypertransmission ≥ 250 μm.

After annotation, the images with annotation labels were exported from Labelme in png format with raw images, label masks, and combined masks, which were further used to develop the CNN models. Python (3.8.5) scripts were applied to extract each label mask and generate regarding masks respectively.

### 2.4. Model Development

The models were trained on a workstation equipped with a 12 Gb NVIDIA GTX 3080 graphics card, utilising PyTorch and TensorFlow deep learning frameworks, and version 2.12.0 Unet architecture, inspired by Ronneberger’s work in 2015 [20], was applied for this semantic segmentation. Unet is a cutting-edge algorithm for semantic segmentation. It is based on fully convolutional networks using encoder-decoder network architecture [26]. The architecture was introduced by Olaf Ronneberger and his team in 2015 [20]. The architecture of Unet looks like the shape of “U”, which defines its name. This architecture consists of a contracting path to capture context and a symmetric expanding path that enables precise localization [20]. Each path includes many blocks, such as the convolutional layer, max pooling, and up-sampling layer.

The model development included first developing a 2D Unet model using single OCT slices, followed by 3D Unet model development with volumetric OCT sections and model evaluation (Figure 1). The initial model development process, with the implementation of a 2D Unet model, leveraged its efficiency in processing individual slices of data. This was then adapted to a 3D Unet model, which allows for volumetric analysis by considering the spatial context across consecutive slices, offering a more comprehensive segmentation. The evaluation of both models was performed using the Dice similarity coefficient (DSC), Precision, Recall, and F1 on a one-dimensional level, as illustrated in Figure 1.

#### 2.4.1. 2D Unet Model Development

4466 annotated images were randomly split into three datasets at patient level: a training dataset (0.8, 3617 images from 100 eyes in 72 patients), a validation dataset (0.1, 402 images from 12 eyes in 8 patients), and a testing dataset (0.1, 447 images from 13 eyes in 9 patients). Images were processed by padding and resizing to 256 × 256 pixels as the input. 

Data augmentation was performed on the training images by random rotation (angle between −10° and 10°), horizontal and vertical flips, and increasing blur, noise, and contrast based on RandAugment [27]. Data augmentation can artificially increase the size of training datasets to improve the model’s performance and prevent model overfitting overall.

After preparation, raw images with masks were fed into a 2D Unet network, as previously described [22], using a Pytorch framework (1.9.1) in a Python environment based on Anaconda software (2.3.2). Each model was trained for 100 epochs with a batch size of 16, and parameters were optimized using the Adam optimizer proposed in 2015 [28]. 

Initially, we trained six 2D Unet models separately, based on individual annotation labels: (i) model of interruption of outer retina, (ii) model of interruption of RPE, (iii) model of absence of outer retina, (iv) model of absence of RPE, (v) model of hypertransmission < 250 μm, and (vi) model of hypertransmission ≥ 250 μm. Next, we trained an automated multiple-label model and ensured that every pixel was uniquely classified into one of the six regions of interest with no overlapping.

#### 2.4.2. 3D Unet Model Development

Pretrained 2D models were applied to all 3D data and generated probability predictions as new masks. In order to fit the 3D model input, 15 black slices (256 × 256 pixels) were added into each volume. Finally, 1241 volumetric OCT images and labels were prepared for further 3D model training with a standard size of 256 × 256 × 64 in each volume. Next, 3D volumetric data were randomly split into two datasets at the patient level: 1110 volumes (0.9) from 113 eyes in 81 patients for training, and 131 volumes (0.1) from 12 eyes in 8 patients for testing. During the training, 5-fold cross-validation was applied in the training dataset. The 1110 volumes were split into 5 separate folds at patient level, and the models were trained on regarding 4 folds while the 1 remaining fold was used for validation. This procedure was repeated 5 times during the entire training period.

3D segmentation models from a Python library were applied [29], which were based on the TensorFlow framework(2.12.0). Each model was trained for 100 epochs with a batch size of 8 and a patch size of 64. Detailed parameters were listed below: “imagenet” as the encoder weights, “vgg16” as the backbone, “softmax” as the activation, and “Adam” as the optimizer.

Similar to the 2D Unet models, we trained six separate atrophy-associated models and one multiclass model.

### 2.5. Model Evaluation

The Dice similarity coefficient (DSC), Precision, Recall, and F1 scores were calculated to evaluate the models’ performance on a one-dimensional level.

The DSC score was the primary outcome to evaluate the models. DSC is a spatial overlapping index for semantic segmentation, which is used to calculate the overlapping proportion of the ground truth and the prediction [30]. A DSC score ranges from 0 to 1, with 0 indicating no overlapping area and 1 indicating a fully overlapping area. The formula is as follows:DSC = (2 × Area of Overlapping)/Total area covering the annotated and predicted pixel regions

Precision, also called the positive predictive value (PPV), shows the ability to predict true positives from all the positives [31]. The formula is as follows:Precision = TP/(TP + FP) (TP: True Positive; FP: False Positive)

Recall, also known as Sensitivity, or the true positive value (TPV), shows the ability to detect a true positive from all the predictions [31]. The formula is as follows:Sensitivity = TP/(TP + FN) (TP: True Positive; FN: False Negative)

The F1 Score is a measure of the model’s performance that combines and balances the Precision and Recall metrics into one single metric and can be used for imbalanced data. In multi-label classification, the F1 score is commonly used to evaluate the classification performance [32]. It is a comprehensive evaluating indicator for machine learning; the higher the score, the better the model’s performance. The formula is as follows:F1 score = 2 × P × R/(P + R) (P: Precision; R: Recall)

### 2.6. Learning Curve Analysis

To determine if the sample size was sufficient to train the robust model, we performed a learning curve analysis. The training images or volumes were randomly separated by setting different percentages (5%, 10%, 15%, etc.). Furthermore, the DSC performance was evaluated on the testing datasets to compare the DSC performance of each training model.

## 3. Results 

### 3.1. Summary of Manual Annotation 

Six atrophy-associated features were manually annotated by two graders using different colors (Figure 2). In total, 19,021 labels were annotated on 4466 OCT images (Table 1).

### 3.2. Performance of Models

#### 3.2.1. Comparison Between the 2D and 3D Models

Overall, the 3D models demonstrated stronger performance, according to the evaluations of the DSC, Precision, Recall and F1 score, compared to the 2D models (Table 2).

The 3D models showed better performance according to the DSC score in all aspects of atrophy features (0.90 ± 0.14, 0.95 ± 0.03, 0.95 ± 0.08, 0.94 ± 0.08, 0.77 ± 0.21, 0.77 ± 0.22, and 0.79 ± 0.20 in the multiclass model, relating to the absence of EZ, absence of RPE, hypertransmission ≥ 250 μm, hypertransmission < 250 μm, interruption of EZ, and interruption of RPE models, respectively), compared to the 2D models (0.87 ± 0.15, 0.94 ± 0.10, 0.90 ± 0.12, 0.88 ± 0.16, 0.63 ± 0.20, 0.65 ± 0.20, and 0.77 ± 0.20 in the multiclass model, relating to the absence of EZ, absence of RPE, hypertransmission ≥ 250 μm, hypertransmission < 250 μm, interruption of EZ, and interruption of RPE model, respectively) (Figure 3). In addition, there was a significant statistical difference in terms of the hypertransmission and interruption of EZ (*p* < 0.001), showing that the 3D models dramatically enhanced the ability to detect these lesions when compared to the 2D models.

Similarly, in terms of the Precision and Recall scores, the 3D models were more strongly and efficiently validated when compared to the 2D models (Figure 4 and Figure 5).

The F1 score is a reliable evaluation metric that combines the Precision and Recall scores and provides a comprehensive evaluation of prediction skills overcoming class imbalance. Additionally, the 3D models indicated robust performance compared to the 2D models (*p* < 0.001), especially in the models of absence of RPE, hypertransmission ≥ 250 μm, hypertransmission < 250 μm, and interruption of EZ (Figure 6).

In addition, detailed results from each round of cross-validation were summarized in Appendix A.

#### 3.2.2. Model Performances Compared to Human Graders

When compared to the same testing dataset during the stage of 2D training, the average DSC values of the 2D model were lower than the average DSC scores between the human graders (grader 1 WW as the ground truth), indicating a much better performance in all aspects (Table 3), with significant statistical differences (*p* < 0.001, *p* < 0.001, *p* < 0.001, *p* < 0.001, *p* = 0.25, *p* = 0.01, and *p* < 0.001, respectively) (Figure 7). Disagreements between the graders varied widely, even after standard training in some features of atrophy, particularly hypertransmission and interruption, which are difficult to define in real-world datasets. However, compared to the human graders, the prediction by models provided relatively steady results and showed less deviation when assessing these features.

#### 3.2.3. Learning Curve Analysis of the 3D Model

As expected, the DSC performance (multiclass) increased with the sample size, plateauing when the percentage of the training dataset reached 85%, with a performance close to 0.89, which means the sample size for the 3D model was sufficient. However, there is need for further improvement of other elements for a better performance, such as the backbone, batch size and epoch (Figure 8). 

Before 3D model development, we also did learning curve of 2D model, and found the sample size was not sufficient eventually (Appendix A). Therefore, data augmentation was applied to overcome this shortage.

## 4. Discussion

This is the first application of a 3D Unet framework to detect all six features of atrophy, and this method that combines 2D and 3D Unet together is unique and novel. Overall, our results are very encouraging, being reliable and reproducible. The currently published literature describes models that can only quantitatively detect features of atrophy that are based mostly on the late stage, using 2D Unet [23,24,25]. We have successfully developed this novel automated method to detect atrophy at an early stage, which is fully invisible from conventional imaging devices, such as CFP, FAF and NIR. Furthermore, this early detection provides a potential extended window for treatments and can even objectively assess the effectiveness of new drugs through the quantitative measurement of expanding atrophy over time. In addition, atrophy-associated features predicted by our models were superior to human graders overall.

We have previously developed a method to detect these six atrophic features, based on 2D Unet [22], with promising results. However, this model did not perform as well as expected for certain features (hypertransmisstion < 250 µm and interruption of RPE and EZ), possibly due to the 2D model framework. OCT provides volumetric three-dimensional imaging information from patients’ retinas, but a 2D Unet model only provides two-dimensional information and probably misses some spatial correlations between structural features or lesions. Therefore, we applied 3D Unet architecture in this study, which has been validated as a deep learning framework that is more suitable for OCT images and performed better than 2D Unet architecture.

Furthermore, we improved the methods for our manual annotation by outlining the boundary of hypertransmission instead of the width, as has been described previously [22]. The reason for this change is to attempt to increase the pixel counts as the input for model training to improve the performance, and the results were in line with our expectations. The performance on both hypertransmission ≥ 250 μm and hypertransmission < 250 μm was better than in our previous study. However, the label of hypertransmission < 250 μm still failed to attain the ideal prediction we expected. Therefore, there is a need to optimize our annotation in order to further decrease human bias. 

We use data augmentation here because our sample size at the initial stage was not sufficient compared to the other study [24]. Data augmentation can artificially increase the amount of training datasets to improve model performance and prevent model overfitting [33]. This augmentation has been validated effectively in our 2D model training, gaining promising results in each label during this initial stage.

Quantified biomarkers in OCT are more likely to elucidate the structure–function correlation in MA in AMD and further the understanding of pathophysiological mechanisms in disease development and progression [34]. Our quantified model prediction is therefore able to monitor MA progression in the longer term, allowing not only the assessment of the effectiveness of treatments as previously mentioned but also offering the potential to slow the progression and monitor this change as new promising drugs for MA such as pegcetacoplan (C3 inhibitor) and avacincaptad pegol (C5 inhibitor) [35] are developed. With the FDA approving the first of these drugs in 2023, there is currently more potential for treatment to stabilize the vision of these patients and finding these early biomarkers may have a significant impact. 

Furthermore, there is the potential to find more biomarkers in OCT scans associated with MA progression, which can more robustly identify the death of the photoreceptors, RPE, and choriocapillaris, which accompany irreversible and permanent vision loss [36,37].

A further application of our models may be in training junior doctors. The biggest challenge to overcome is to accurately evaluate data from the real world that may have poor image quality. This is mainly due to the difficulties encountered in capturing high resolution images of the retina in AMD patients, who are usually over the age of 50 years and may suffer from a cataract, which interferes with visualization of the retina. Despite using two experienced graders, who have been ophthalmologists for over 7 years, to perform manual annotation, we observed variable disagreements and human bias. Therefore, there would be challenges for less experienced junior doctors to detect the lesions correctly, and our models provide the potential for training these young doctors and shortening their learning curve to become experienced imaging readers.

However, there are also several limitations to this study. The biggest limitation is the difference in some labels’ performance compared to others, particularly the label of hypertransmission < 250 μm. It is difficult to outline this tiny lesion, especially when the resolution is poor. Therefore, its performance is slightly worse than that seen by human graders. In addition, the labels of interruption of RPE and the outer retina showed a less significant difference in performance than the other labels, although their performance was still superior to human graders. This is likely due to the fact that labeling this type of small change in interruption can be difficult, particularly to delineate the outline accurately. Inevitably, it often generates human bias, and our human graders failed to get good agreement on these labels. 

3D deep learning networks often require a large number of training datasets and parameters, especially for volumetric medical images, as the depth of the image volume varies from 20 to 400 slices per scan, which contain specific information about lesions, such as their size, location and correlation to surrounding tissues [21]. Usually, high resolution scan volumes are of the size 512 × 512, and need to be down sampled before being fed into the 3D network due to the availability of computer memory [21]. In order to reduce the computational cost, researchers usually have to reduce and resize the overall size of medical images and volumes at the cost of significant information loss [21]. Balancing the resizing of volumetric medical images without the loss of significant information [21] is still a major challenge, requiring further work to explore this area in the future. 

Another limitation is that our results sometimes show slight deviation and are not stable. The main reason for this is the image quality. The data we used in this study is all from the real world; therefore, the quality of the images varies and cannot be kept stable, although we did exclude images with bad quality. We will address this limitation and set up an even stricter standard of inclusion criteria for images in the future. 

Another limitation is that the images we used were all from the same OCT machine, and we need further external validation of our model using other OCT machines. In addition to generating an even more powerful and generic AI model, more patients, of different races and from multi-centers, need to be involved in the future. 

## 5. Conclusions

In summary, we have successfully developed a combination of 2D and 3D Unet frameworks to detect all six atrophy-associated features in patients with AMD at an early stage, with encouraging and promising results. The reliability and reproducibility of our results underscore the potential of this technology to enhance clinical decision-making and patient care. However, addressing the identified limitations and pursuing further validation will be crucial to realizing the full clinical potential of our approach.

## Figures and Tables

**Figure 1 bioengineering-11-01191-f001:**
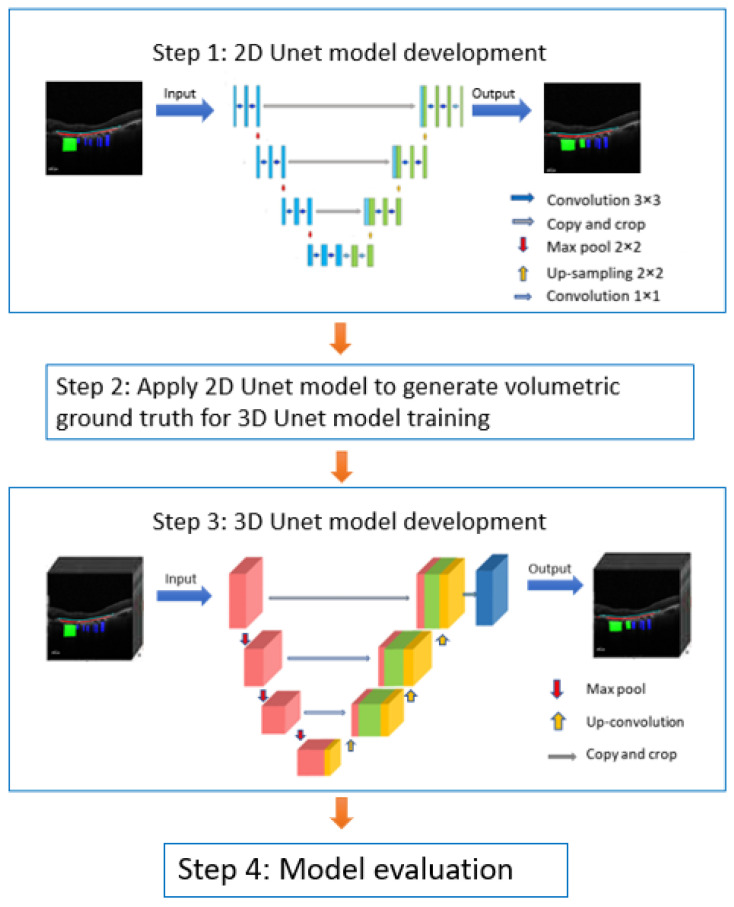
Workflow of model development. Firstly, 4466 raw images with manual annotations were fed into a 2D Unet framework for model development. Next, the 2D Unet model was applied to generate volumetric ground truth from 1241 OCT scans, followed by 3D Unet model development. Finally, the 2D and 3D models were comprehensively evaluated using DSC, Precision, Recall, and F1 scores.

**Figure 2 bioengineering-11-01191-f002:**
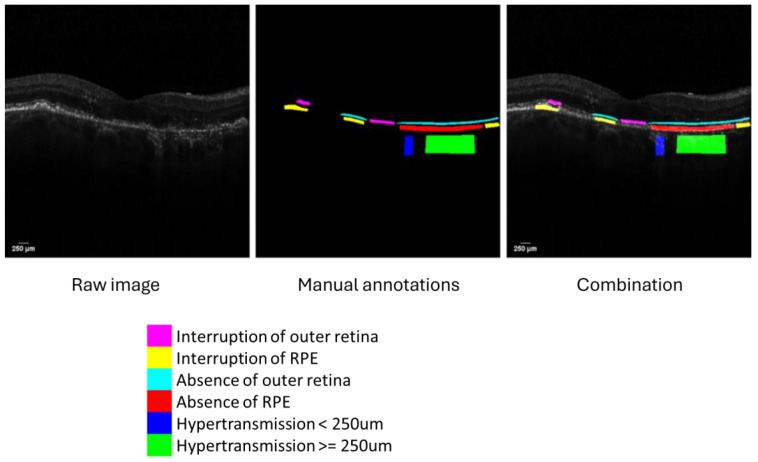
Example of manual annotation. A total of 6 different atrophy-associated features were manually annotated by 2 graders using different colors.

**Figure 3 bioengineering-11-01191-f003:**
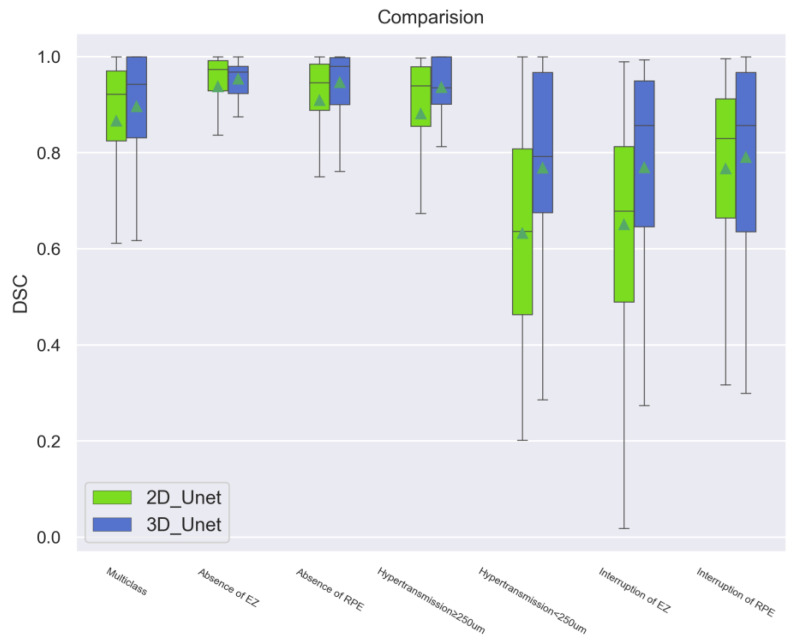
Comparison of the DSC scores between the 2D and 3D models. The 3D models showed steady superiority, both in the multiclass model and in each independent model, compared to the 2D models. In addition, the 3D models showed remarkable results in terms of Hypertransmission and Interruption of EZ, with significant statistical differences (*p* < 0.001).

**Figure 4 bioengineering-11-01191-f004:**
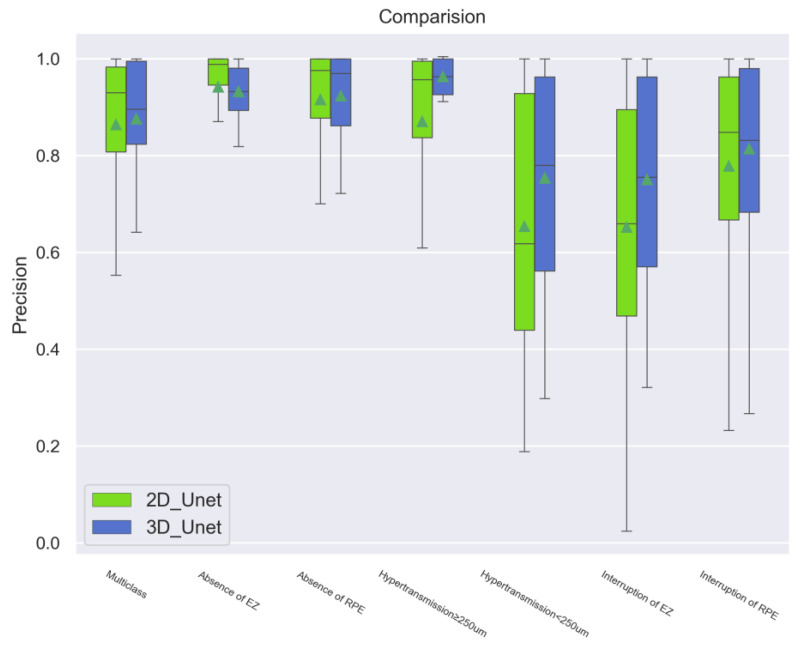
Comparison of the Precision scores between the 2D and 3D models. Overall, the 3D models performed strongly in all aspects measuring the accuracy of the quantity of “positive” predictions made by the models.

**Figure 5 bioengineering-11-01191-f005:**
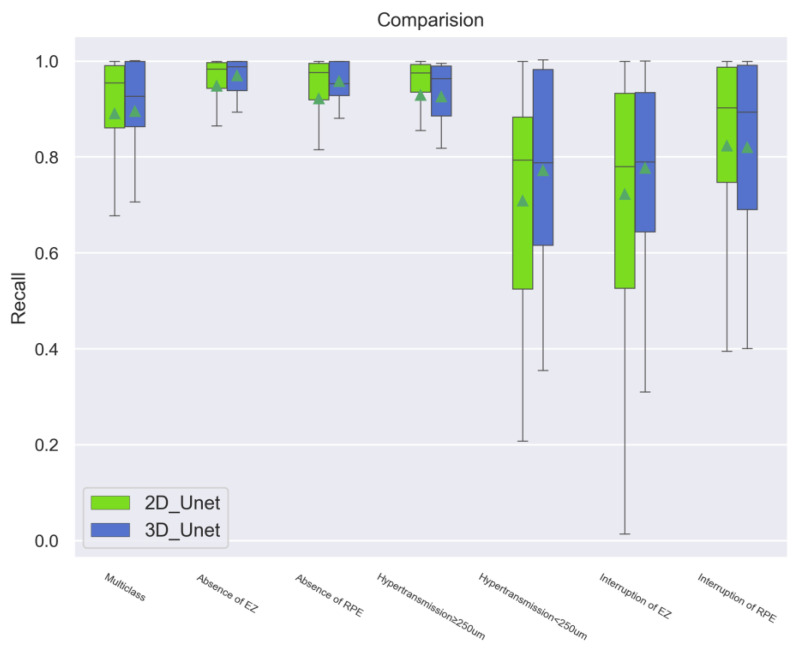
Comparison of the Recall scores between the 2D and 3D models. Similar to the performance of Precision, the 3D models showed a steady performance compared to the 2D models, generally measuring the accuracy of the quantity of “positive” samples in the datasets identified by the models correctly.

**Figure 6 bioengineering-11-01191-f006:**
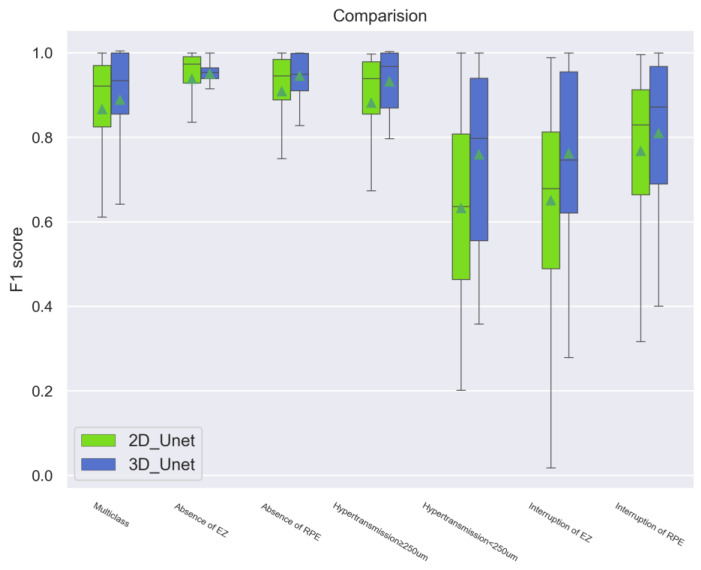
Comparison of the F1 score between the 2D and 3D models. The 3D models indicated a robust performance compared to the 2D models, especially in the models of absence of RPE, hypertransmission ≥ 250 μm, hypertransmission < 250 μm, and interruption of EZ, which demonstrated significant statistical differences (*p* < 0.001). In summary, the 3D models showed a reliable and reproducible ability to detect atrophy.

**Figure 7 bioengineering-11-01191-f007:**
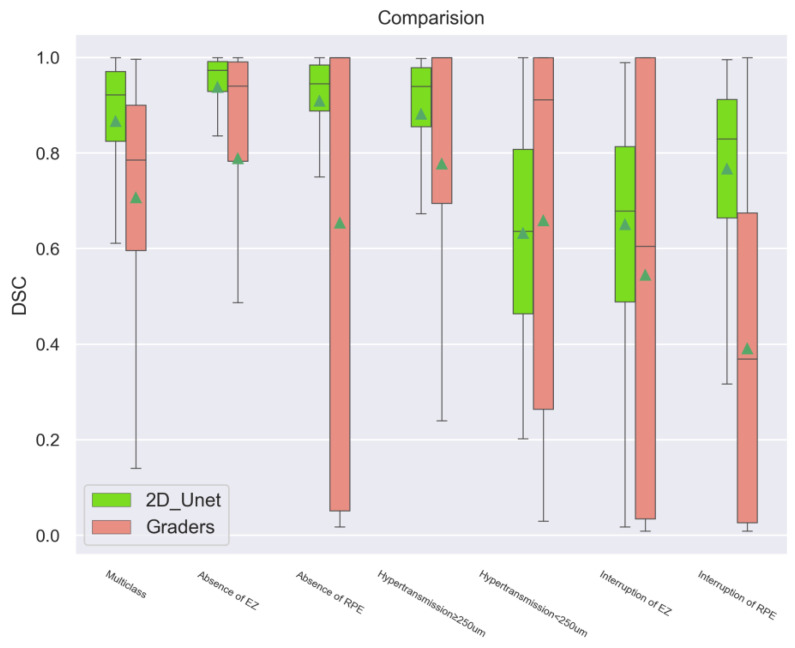
Comparison between the human graders and the 2D Unet model. Generally, the 2D models showed a stronger overlapping ratio than the human graders when grader 1 was used as the ground truth. However, it is noticeable that there was no significant difference in the performance of hypertransmission < 250 μm, probably due to the difficulty of the manual annotation itself.

**Figure 8 bioengineering-11-01191-f008:**
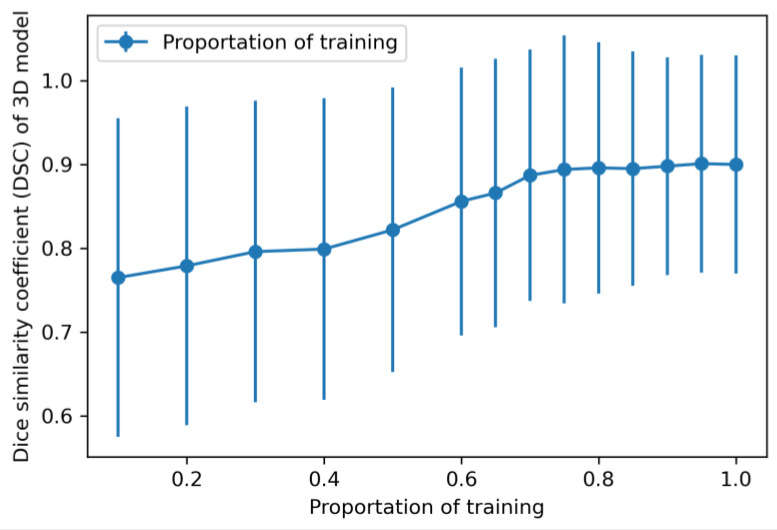
Learning curve of 3D model training. Overall, the DSC score elevated gradually with the enlargement of sample size. When the sample size reached 85% of the entire training dataset, the performance plateaued with a DSC score close to 0.89, which showed that the sample size for the 3D model training was sufficient.

**Table 1 bioengineering-11-01191-t001:** Summary of each label annotated.

Label	Numbers of Each Label	Pixel Counts
Interruption of outer retina	2425 (12.75%)	1,132,765 (0.10%)
Interruption of RPE	3510 (18.45%)	2,413,401 (0.21%)
Absence of outer retina	3569 (18.76%)	4,291,444 (0.38%)
Absence of RPE	2390 (12.57%)	3,168,831 (0.28%)
Hypertransmission < 250 μm	3930 (20.66%)	2,176,126 (0.19%)
Hypertransmission ≥ 250 μm	3197 (16.81%)	20,127,892 (1.77%)
Total	19,021	1,134,149,632 (including background)

**Table 2 bioengineering-11-01191-t002:** Comparison of performance between the 2D and 3D models.

**Multiclass**	**One-Dimensional evaluation**							
	**DSC**		**Precision**		**Recall**		**F1 Score**	
2D_Unet	0.87 ± 0.15		0.86 ± 0.17		0.90 ± 0.16		0.87 ± 0.15	
3D_Unet	0.90 ± 0.14		0.88 ± 0.14		0.90 ± 0.15		0.89 ± 0.14	
*p* value	0.07		0.51		0.78		0.19	
**Absence of EZ**	**One-dimensional evaluation**							
	DSC		Precision		Recall		F1 score	
2D_Unet	0.94 ± 0.10		0.94 ± 0.11		0.95 ± 0.10		0.94 ± 0.10	
3D_Unet	0.95 ± 0.03		0.93 ± 0.06		0.97 ± 0.03		0.95 ± 0.02	
*p* value	0.14		0.41		0.03		0.26	
**Absence of RPE**	**One-dimensional evaluation**							
	DSC		Precision		Recall		F1 score	
2D_Unet	0.90 ± 0.12		0.92 ± 0.13		0.92 ± 0.14		0.91 ± 0.12	
3D_Unet	0.95 ± 0.08		0.92 ± 0.11		0.96 ± 0.05		0.95 ± 0.07	
*p* value	<0.001		0.58		0.01		<0.001	
**Hypertransmission ≥ 250 μm**		**One-dimensional evaluation**						
		DSC		Precision		Recall		F1 score
2D_Unet		0.88 ± 0.16		0.87 ± 0.20		0.93 ± 0.13		0.88 ± 0.16
3D_Unet		0.94 ± 0.08		0.96 ± 0.06		0.93 ± 0.11		0.93 ± 0.08
*p* value		<0.001		<0.001		0.81		<0.001
**Hypertransmission < 250 μm**		**One-dimensional evaluation**						
		DSC		Precision		Recall		F1 score
2D_Unet		0.63 ± 0.20		0.65 ± 0.26		0.71 ± 0.24		0.63 ± 0.20
3D_Unet		0.77 ± 0.21		0.75 ± 0.25		0.77 ± 0.20		0.76 ± 0.20
*p* value		<0.001		0.01		0.05		<0.001
**Interruption of EZ**		**One-dimensional evaluation**						
		DSC		Precision		Recall		F1 score
2D_Unet		0.65 ± 0.20		0.65 ± 0.25		0.72 ± 0.24		0.65 ± 0.20
3D_Unet		0.77 ± 0.22		0.75 ± 0.24		0.78 ± 0.19		0.76 ± 0.19
*p* value		<0.001		<0.001		0.06		<0.001
**Interruption of RPE**		**One-dimensional evaluation**						
		DSC		Precision		Recall		F1 score
2D_Unet		0.77 ± 0.20		0.78 ± 0.22		0.82 ± 0.22		0.77 ± 0.20
3D_Unet		0.79 ± 0.20		0.81 ± 0.19		0.82 ± 0.21		0.81 ± 0.20
*p* value		0.34		0.18		0.92		0.09

**Table 3 bioengineering-11-01191-t003:** Comparison between the 2 graders (grader 1 as the ground truth).

Name	DSC Score
Multiclass	0.71 ± 0.27
Absence of EZ	0.79 ± 0.32
Absence of RPE	0.65 ± 0.46
Hypertransmission ≥ 250 μm	0.78 ± 0.36
Hypertransmission < 250 μm	0.66 ± 0.37
Interruption of EZ	0.55 ± 0.44
Interruption of RPE	0.39 ± 0.33

## Data Availability

The real-life clinical datasets used in the current study are not publicly available due to privacy constraints. The data can be requested for sharing for peer-review or research purposes by contacting Wei Wei (w.wei20@imperial.ac.uk).

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
