# Peer review of "Early Detection of Macular Atrophy Automated Through 2D and 3D Unet Deep Learning"

_bioengineering, 2024, doi:10.3390/bioengineering11121191_

Round 1
Reviewer 1 Report
Comments and Suggestions for Authors
Congratulations to the authors for this interesting work on geographic atrophy
I have nothing to add
Author Response
thanks very much for your feedback.
Reviewer 2 Report
Comments and Suggestions for Authors
The paper by Wei et al describes application of 2D and 3D Unet for the detection of macular atrophy. Theme of the study is important and interesting. The authors provided quite full description of the proposed approach and achived results. At the same time, there is still some major issues that must be addressed before the publication.
1. The achived perfomances sometimes demonstrates values more than 1 (eg table 1 line 1 "0.86±0.17 0.90±0.16"). Does this indicate the insufficiency of the proposed training approach?
2. Some presented p-values = 0.00. Does this means p<0.01?
3. Results are missing for validation and training. The data is presented only for testing, and only text explanations sometimes appear for training and test.
4. Some evidence is required to prove that 100 epochs provides the best model training.
5. SD data must be added to figure 8.
6. Discussion is missing comparison to other approches for OCT images segmentation and classification.
7. Figure 1 is confusing. There is no explanation of layers types. The output is the same as the input. There is no training/validation/testing.
8. It is not clear from the text how the authors may confirm that the analyzed patients have macular atrophy at the early stage.
The paper may be published after correction of mentioned issues.
Reviewer 3 Report
Comments and Suggestions for Authors
Lines 93-95: there are certain conditions that can affect the quality of the OCT image and still pass the exclusion criteria (eg. corneal mild haze, mild subcapsular cataract, vitreous degenerative conditions). Please add these to the exclusion criteria.
Your model is very interesting, but to validate it, further work is required: much more eyes to be tested, different OCT machines, several research centers. Please comment.
How does your method influence the outcome of the geographical atrophy, as complement inhibitors are (at least for now) on the controversial side?
Round 2
Reviewer 2 Report
Comments and Suggestions for Authors
The authors addressed arised issues, the paper may be published.